# Seed Transmission of Wheat Streak Mosaic Virus and Triticum Mosaic Virus in Differentially Resistant Wheat Cultivars

**DOI:** 10.3390/v15081774

**Published:** 2023-08-21

**Authors:** Saurabh Gautam, Senthilraja Chinnaiah, Benjamin Herron, Fekede Workneh, Charles M. Rush, Kiran R. Gadhave

**Affiliations:** 1Texas A & M AgriLife Research, 6500 W Amarillo Blvd, Amarillo, TX 79106, USA; saurabh.gautam@agnet.tamu.edu (S.G.); s.chinnaiah@ag.tamu.edu (S.C.); benjamin.herron@ag.tamu.edu (B.H.); fekede.workneh@ag.tamu.edu (F.W.); charlie.rush@ag.tamu.edu (C.M.R.); 2Department of Entomology, Texas A & M University, College Station, TX 77843, USA; 3Department of Plant Pathology, Texas A & M University, College Station, TX 77840, USA

**Keywords:** *Tritimovirus*, *Poacevirus*, *Potyviridae*, virus–virus interactions, seed viruses

## Abstract

Wheat streak mosaic virus (WSMV) and Triticum mosaic virus (TriMV) are important viral pathogens of wheat in the Great Plains. These viruses individually or in mixed infections with High Plains wheat mosaic virus cause a devastating wheat streak mosaic (WSM) disease. Although seed transmission of WSMV has been studied, no information is currently available on that of TriMV. Furthermore, no study has explored the implications of mixed infections of WSMV and TriMV on seed transmission of one or both viruses. To study both aspects, seeds from differentially resistant field-grown wheat plants (cv. TAM 304 (susceptible), Joe (WSMV resistant, *Wsm2* gene), and Breakthrough (BT) (WSMV and TriMV resistant, *Wsm1* gene)) showing characteristic WSM symptoms were collected and analyzed to quantify both viruses using qRT-PCR. The percentage of seeds tested positive for WSMV or TriMV individually and in mixed infection varied with cultivar and virus combinations; 13% of TAM 304 seeds tested positive for WSMV, followed by 8% of BT and 4% of Joe seeds. Similarly, TriMV was detected in 12% of BT seeds, followed by 11% of TAM 304 and 8% of Joe seeds. Lastly, mixed infection was detected in 7% of TAM 304 seeds, followed by 4% in BT, and 2% in Joe. Dissection of field-collected seeds into three parts, embryo, endosperm, and seed coat, revealed both WSMV and TriMV accumulated only in the seed coat. Consistent with seeds, percent infection of WSMV or TriMV in the plants that emerged from infected seeds in each treatment varied with cultivar and virus combinations (WSMV: BT 3%; Joe 2%; TAM 304 9%; TriMV: BT 7%; Joe 8%; and TAM 304 10%). Plants infected with mixed viruses showed more pronounced WSM symptoms compared to individual infections. However, both viruses were present only in a few plants (BT: 2%, Joe: 1%, and TAM 304: 4%). Taken together, this study showed that TriMV was transmitted vertically at a higher frequency than WSMV in resistant cultivars, and the seed transmission of TriMV with WSMV increased the virulence of both pathogens (measured via WSM symptom severity) in the emerged plants. Furthermore, *Wsm1* and *Wsm2* genes considerably reduced WSMV transmission via infected seeds. However, no such effects were observed on TriMV, especially in progeny plants. These results reiterated the importance of planting clean seeds and highlighted the immediate need to identify/develop new sources of TriMV resistance to effectively manage the recurring WSM epidemic.

## 1. Introduction

Wheat (*Triticum aestivum* L.) is one of the major staple food crops worldwide. As the third largest producer of wheat in the US, Texas wheat production totaled about 34 million bushels in 2016–2017, generating approximately $255 million in revenue [1]. In the Texas Panhandle region and across the entire Great Plains, hard winter wheat (planted late August–November) is a dual-purpose crop; it provides forage for the cattle industry in late fall and winter, and it is harvested for grains in mid-summer. In this region, the wheat curl mite (WCM), *Aceria tosichella* Keifer (Acari: *Eriophyidae*), and WCM-transmitted wheat streak mosaic (WSM) are major constraints to winter wheat production, as they significantly impact grain yields [2,3]. Three major pathogens causing WSM are wheat streak mosaic virus (WSMV, *Tritimovirus wheat streak mosaic virus,* and *Potyviridae*), Triticum mosaic virus (TriMV, *Poacevirus Triticum mosaic virus*, *and Potyviridae*), and High Plains wheat mosaic virus (HPWMoV, *Emaravirus High Plains wheat mosaic virus*, and *Fimoviridae*) [4,5]. Mixed infection of these viruses produces indistinguishable symptoms (yellow, mosaic-like streaks on the leaves) compared to their individual infections. TriMV often occurs in mixed infection with WSMV and has synergistic effects in increasing disease severity and yield losses [6,7,8]. Historically, out of these viruses, WSMV was most widely distributed across Texas and the American Great Plains [9]. However, recently, TriMV is becoming increasingly economically important possibly due to the lack of genetically resistant wheat cultivars [9]. On the contrary, HPWMoV is rarely found in the Panhandle [9]. WSMV and TriMV are filamentous viruses with positive-sense, single-stranded RNA genomes [10,11,12]. WSMV was reported over a century ago in 1922 in Nebraska, USA [13], whereas TriMV was first documented in Kansas in 2006 [14].

Management of WSMV and TriMV is focused on planting resistant cultivars and disrupting the virus transmission cycle by removing ‘green bridge’ summer alternate hosts for WCM and associated viruses [15]. Currently, *Wsm1* and *Wsm2* are the two most widely used resistance genes [16,17] that have been introgressed into commercial wheat cultivars from wheatgrass (*Thinopyrum intermedium* (Host) Barkworth & D. R. Dewey) and wheat germplasm line CO960293-2, respectively [16,18]. However, resistance offered by these single-dominant genes is temperature dependent and becomes ineffective above 18 °C [16,19,20,21,22]. *Wsm1* has been reported to offer protection against both WSMV and TriMV, whereas *Wsm2* offers protection only against WSMV [23,24].

Transmission of WSMV and TriMV mostly occurs through WCM. Although vector transmission plays a significant role in WSM epidemics in fields, it does not support the season-to-season survival of viruses under unfavorable conditions. On the other hand, seed transmission provides means for plant viruses to survive in a protected seed environment over a long period of time and to establish early infection, which then typically spreads via insect vectors throughout the field [25]. Previous studies reported genotype-dependent low-frequency seed transmission of WSMV [26,27,28]. Seed transmission has been reported to facilitate the intercontinental spread of WSMV in the past and to serve as a reservoir of primary inoculum in the areas where WSMV is prevalent [26]. However, to date, no study has reported on seed transmission of TriMV and its impact on WSMV seed transmission and disease dynamics. Due to the current lack of resistant cultivars that offer comprehensive protection against TriMV, understanding the ever-changing WCM-mediated disease epidemics is important to devise sound pest- and disease-management strategies. We, therefore, studied the incidence of WSMV and TriMV in resistant and non-resistant wheat cultivars at the critically important grain-filling stage of wheat. We further determined the frequency of WSMV/TriMV seed transmission in the mixed-infected seeds collected from the field and assessed how WSMV single-gene resistance (*Wsm1* and *Wsm2*) impacts disease severity in the plants infected vertically via seeds.

## 2. Materials and Methods

### 2.1. Plant Materials, Virus Isolates, and Seed Collection

Two single-gene differentially resistant wheat cultivars, Breakthrough (BT, *Wsm1* gene) and Joe (*Wsm2* gene), and one susceptible cultivar, TAM 304, were used in this study. The field experiment was conducted in 2021–2022 in Bushland, TX, under a center-pivot irrigation system. A strip of TAM 304 (9 m wide) was planted along the south edge of the field site in late July 2021 to serve as a trap crop for WCM and a point source of natural WSMV and TriMV infection for the experiment plots, which were planted in September 2021. Immediately after WSM symptoms appeared (mid-March 2022) in experiment plots, symptomatic plants were flagged (30 for each cultivar) and tested for WSMV, TriMV, and HPWMoV infection via qRT-PCR analysis [8,9]. The qRT-PCR reaction mix contained 1× TaqMan Fast Advanced Master Mix (Applied Biosystems), 0.3 µM forward primer, 0.3 µM reverse primer, and 0.25 µM TaqMan probe. Every PCR reaction was carried out with three negative control samples (wheat plants grown in a lab from seed collected from clean plants grown in a greenhouse) run in duplicates. Field samples with C*t* (cycle threshold) value with at least 3 below the average of negative control (~35) were considered positive. Samples from field plants that tested positive for both WSMV and TriMV were collected over six time points every week, starting March 23rd until the wheat started senescing. Fifteen–twenty plants from each cultivar were hand-harvested for grains 38 weeks after planting on 12th June 2022. Seeds from non-symptomatic plants that tested negative for WSMV, TriMV, and HPWMoV in PCR analysis were used as negative controls. The collected seeds were divided into three lots. Each lot contained seeds from 5–7 infected plants (hereafter referred to as seed lots). All seeds collected from the field were temporarily stored at 4 °C prior to planting. The coat protein of WSMV and TriMV from three randomly selected plants from each cultivar were sequenced using the primers WSMV (WSMVCPF 5′-CCGGATTTCAAGTTGCCCTC-3′ and WSMVCPR-5′-TTAGTACCCGCACTCAGTCG-3′) and TriMV (TriMVCPF 5′-TGGGGAGAACGAAATGGTCA-3′ and TriMVCPR 5′-TAGCTCAGCCTCTTACAAGC-3′). The identities of the obtained CP sequences (TriMV (OQ866321-OQ866329) and WSMV (OQ866330-OQ866338)) were further confirmed via the NCBI Basic Local Alignment Search Tool (BLAST) (https://blast.ncbi.nlm.nih.gov/Blast.cgi, accessed on 14 April 2023).

### 2.2. Phylogenetic Analysis

Coat protein sequences of WSMV and TriMV from Bushland isolates in this study, along with the corresponding WSMV/TriMV CP reference sequences from GenBank, were used for phylogenetic analysis. Geneious Prime was used to retrieve the CP open reading frame from downloaded sequences [29]. For phylogenetic analysis, the nucleotide substitution models for evolutionary analysis were determined using the Phangorn package in R [30]. The best-fitting model (general time-reversible model) was selected based on the Akaike Information Criterion (AIC). A maximum likelihood (ML) phylogenetic tree was generated in Phangorn in R, and node support was assessed using 1000 bootstraps. The obtained phylogenetic tree was visualized in the Interactive Tree of Life software (version 4) [31].

### 2.3. Detection of WSMV and TriMV in Seeds

WSMV and TriMV detection in seeds was performed by qPCR using the procedure described above. Prior to RNA extraction, seeds were surface sterilized by immersion and stirring (1500 rpm for 10 min in 10% commercial bleach) followed by two washes (2 min each) in sterile distilled water. After sterilization, seeds were soaked in sterile Fisherbrand™ Premium Microcentrifuge 1.5 mL tubes (Catalog No. 05-408-129; hereafter referred to as centrifuge tubes) containing 200 μL sterile nuclease-free water overnight at 4 °C to facilitate the grinding of seeds prior to RNA extraction. Geno/Grinder^®^ (SPEX SamplePrep, LLC, NJ, USA) was used for seed grinding followed by total RNA extraction using TRIzol^®^ reagent (Thermo Fisher Scientific, Waltham, MA, USA). One-step qRT-PCR was used to detect and quantify WSMV and TriMV in seeds. Absolute virus copy numbers were estimated using the standards containing known copies of qPCR-targeted RNA of WSMV or TriMV. Standards were designed by Custom Applied Biosystems TaqMan Expression assays (Thermo Fisher Scientific, Waltham, MA, USA). Ten-fold serial dilutions containing 10^1^ to 10^11^ targeted RNA copies were prepared using the standards. Each sample was run in duplicate with all appropriate controls. Seeds (50) from each lot belonging to different cultivars were tested for viruses and the experiment was replicated three times (n = 150).

In addition, a subset of 25 seeds per cultivar were cut longitudinally into two equal halves (each half containing an embryo, endosperm, and seed coat) using a heat-sterilized scalpel. The first half of each seed was tested for the presence of viruses using the procedures above, whereas the remaining half was stored at −20 °C. Embryo, endosperm, and the seed coat from the corresponding half of seeds that tested positive for either of the viruses were dissected and tested individually to understand the accumulation of both viruses in different parts of the seed via qRT-PCR.

### 2.4. Evaluation of WSMV and TriMV Transmission from Seeds to Progeny Plant

To investigate the possible seed transmission of WSMV and TriMV to the derived progeny plants, grow-out experiments were conducted. Ninety randomly collected seeds from infected or non-infected plants belonging to each cultivar were planted in three plastic trays (30 seeds/tray). Each tray contained seeds belonging to different seed lots collected from the field. Trays were transferred to a growth chamber (18 ± 1 °C, 14 h L:10 h D). One-week post-germination, total RNA from 25 mg youngest leaf tissue was extracted, the presences of WSMV and TriMV were tested, and absolute virus copy numbers were estimated with appropriate controls, as described above. Six weeks post-germination, the disease phenotype of each individual plant was examined under a white lamp upon uprooting.

### 2.5. Data Analysis

Data analyses were performed in R version 3.4.2 [32]. Data for the presence or absence of viruses in seed, embryo, endosperm, seed coat, and plants were analyzed using the generalized mixed effect model (‘*glmer*’ function), whereas quantitative data on virus accumulation were analyzed using the linear mixed effect model (‘*lmer*’ function) in the ‘lme4’ package with treatments as the fixed effects and replications as the random effects [33]. ANOVA was run using ‘car’ [34,35]. Repeated measures ANOVA was used to compare virus accumulation in field-flagged plants at different time intervals. Statistical differences were considered significant at *p* < 0.05.

## 3. Results

### 3.1. Virus Accumulation in the Field

All WSM symptomatic plants in the field were mixed–infected with WSMV and TriMV. WSMV (*F* = 89.5, df = 1, *p* < 0.001) and TriMV (*F* = 112, df = 1, *p* < 0.001) accumulation in plants significantly changed between the onset of symptoms until harvest. Except for week 4, WSMV accumulation was significantly higher in TAM 304 compared to BT and Joe at all time points (Figure 1A). In BT and Joe, WSMV accumulation increased up to week 4, whereas it varied up to week 5 and started declining thereafter as wheat started senescing in TAM 304 (Figure 1A). TriMV accumulation was significantly higher in TAM 304 at all time points, except for week 5, where no significant difference between any cultivars was observed (Figure 1B). Of the three cultivars, BT had the lowest TriMV at first four timepoints, which then steeply increased in week 5 before a steep decline as wheat senesced (Figure 1B). In general, WSMV and TriMV accumulated in significantly higher levels in the susceptible cultivar TAM 304 than the WSMV-resistant Joe and WSMV and TriMV-resistant BT (Figure 1).

### 3.2. Phylogenetic Analysis

The phylogenetic tree based on maximum-likelihood analysis of the WSMV CP sequences revealed that WSMV Bushland isolates from TAM 304 along with two WSMV recombinant isolates from Kansas (isolate # KM19; NCBI GenBank accession number MW990168) and Nebraska (isolate # NE01_19; NCBI GenBank accession number MW990167) shared a clade with isolates from Europe, suggesting they were more closely related to isolates from Europe than the US (Figure 2A). Conversely, WSMV isolates from BT and Joe were clustered together with US isolates reported from the American Great Plains.

The phylogenetic tree based on maximum-likelihood analysis of the TriMV CP sequences revealed that all TriMV Bushland isolates were closely related to isolates reported from the US (Figure 2B).

### 3.3. Detection of WSMV and TriMV in Field Collected Seeds

The percentage of seeds infected with WSMV (χ2 = 66.08, df = 2, 447, and *p* < 0.001) or TriMV (χ2 = 81.21, df = 2, 447, and *p* < 0.001) between cultivars differed significantly. A significantly higher percentage of TAM 304 seeds tested positive for WSMV than BT and Joe (Figure 3A). Also, a significantly higher percentage of TAM 304 seeds tested positive for TriMV than Joe (Figure 3B) but not BT. WSMV (F = 68.28, df = 2, 38, and *p* < 0.001) and TriMV (F = 28.67, df = 2, 38, and *p* < 0.001) accumulation differed significantly between cultivars. TAM 304 seeds had significantly higher WSMV and TriMV than seeds collected from Joe and BT plants (Figure 3C,D). Furthermore, within resistant cultivars BT accumulated significantly lower WSMV than Joe (Figure 3C). However, TriMV accumulation did not differ between resistant cultivars (Figure 3D).

Of the 150 seeds tested, 10 in TAM 304, 6 in BT, and 4 in Joe were infected with both WSMV and TriMV (χ2 = 8.48, df = 5, 894, *p* = 0.21) (Figure 4A). Compared to TAM304, WSMV (F = 64.27, df = 2, 37, *p* < 0.001) and TriMV (F = 14.07, df = 2, 37, *p* < 0.001) accumulations were significantly lower in singly as well as mixed infections in BT and Joe cultivars. However, within cultivars, the amount of WSMV or TriMV did not differ between mixed- vs. single-infected TAM 304, Joe, or BT seeds (Figure 4B,C).

### 3.4. Distribution of WSMV and TriMV in Infected Seeds

Across all three cultivars, both WSMV and TriMV accumulated in the seed coat but not in the embryo or endosperm. WSMV accumulation in the seed coat differed significantly between cultivars (*F* = 42.37, df = 2, 72, and *p* = 0.02). The seed coat dissected from TAM 304 seeds had significantly higher WSMV than the seed coat dissected from the Joe and BT cultivar seeds (Figure 5A). However, the TriMV accumulation in the seed coat did not differ between cultivars (*F* = 7.27, df = 1, 72, and *p* = 0.57) (Figure 5B).

### 3.5. Transmission of WSMV and TriMV from Seed to Progeny Wheat Plants

In seedlings that emerged from infected seeds, the percentage of infection by WSMV differed significantly between cultivars (*χ2* = 82.11, df = 2, 267, and *p* < 0.001). A significantly higher percentage of TAM 304 and BT plants tested positive for WSMV than Joe (Figure 6A). However, the percentage of infection by TriMV did not differ significantly between cultivars (*χ2* = 12.08, df = 2, 267, *p* = 0.21) (Figure 6B).

The WSMV accumulation level in plants that emerged from infected seeds was dependent on the wheat cultivar (*F* = 39.28, df = 5, 31, and *p* < 0.001). TAM 304 had significantly higher WSMV than BT and Joe (6C). In contrast, TriMV accumulation did not differ between cultivars (*F* = 9.12, df = 5, 42, and *p* = 0.51) (Figure 6D).

Not all plants tested positive for WSMV or TriMV in qPCR-developed disease phenotype. For BT, only 5 of 15 (33.33%) plants that tested positive for either of the viruses developed disease phenotype. In TAM 304, it was 9 of 23 (39.13%), whereas only 3 out of 14 (21.42%) plants that tested positive for either of the viruses developed disease phenotype in Joe. However, all plants tested positive for both WSMV and TriMV in qPCR-developed WSM disease. Mixed infection (WSMV and TriMV) was observed in four plants in TAM 304, in two plants for BT, and in one plant from the cultivar Joe. Furthermore, all mixed-infected plants (TAM 304: 4, BT: 2, and Joe: 1) developed a more severe disease phenotype than plants infected with either WSMV or TriMV (Figure 7).

## 4. Discussion

In the US, hard red winter wheat accounts for about 40% of total production and is grown primarily in the Great Plains [36]. In this region, WSMV and TriMV cause a great deal of losses to wheat production. While mixed infection of both viruses is common, a comprehensive study on seed transmission of them, especially TriMV, is currently lacking. In the current study, we found during critical developmental stages of wheat that WSMV and TriMV titers were significantly higher in the susceptible cultivar ‘TAM 304’ than the resistant cultivars ‘BT’ and ‘Joe’. However, low virus titer did not prevent seed transmission of WSMV and TriMV in resistant wheat cultivars. Furthermore, the significantly higher WSMV and TriMV accumulated in seeds collected from susceptible plants in comparison to seeds collected from resistant plants. However, not all next-generation plants that tested positive for WSMV or TriMV showed phenotypic disease expression. Across the cultivars, 21–39% of plants that tested positive for WSMV or TriMV showed phenotypic disease expression. On the contrary, 100% of plants that had vertical infection with both viruses via seeds developed the WSM disease phenotype. To our knowledge, this is the first comprehensive study on seed transmission of TriMV and WSMV in a mixed-infection scenario, which is most common in wheat-growing regions of the US. This also is the first study to analyze individual wheat seeds to assess the distribution of both viruses in different seed components. Taken together, we speculate that TriMV will most likely be dominant in disease epidemics, if it is not already, in Texas and the American Great Plains, due to shared transmission mechanisms (both vertical and horizontal) of WSMV along with TriMV and the availability of limited resistance cultivars against TriMV.

Phylogenetic analysis based on coat protein sequences showed that although Bushland WSMV isolates were collected from the resistant and susceptible cultivars planted within the same plot, there is enough diversity that caused the grouping of these isolates in separate clusters, perhaps due the existence of WSMV quasispecies populations. On the other hand, the genetic variations in TriMV Bushland isolates were low and in the phylogenetic tree; all isolates, irrespective of the host genetics, grouped together in a single clade. Diversity in the WSMV isolates provided an insight into the potential evolutionary pressures, which may have been imposed by resistant genes in BT and Joe cultivars.

Increased levels of TriMV and WSMV accumulation in field-infected plants observed in the current study coincided with increasing temperatures in weeks following the initial WSM symptoms appearances. Virus accumulation in both resistant and susceptible cultivars showed an expected disease progression pattern where virus accumulation increased in the first four weeks, most likely due to the breakdown of *Wsm1*- and *Wsm2*-mediated resistance above 18 °C [21]. On the contrary, virus accumulation declined in the subsequent weeks, plausibly because wheat reached maturity during that time. Despite this decline, WSMV and TriMV accumulated in seeds in both resistant and susceptible cultivars to varying degrees. We observed significant effects of the presence of resistant genes on the vertical movement of viruses; seeds from resistant BT and Joe had significantly lower accumulation of WSMV and TriMV than those from susceptible TAM304. This was plausibly due to the low early accumulation of both viruses in resistant plants. Despite plausible resistance breaking at high temperatures at grain formation and filling stages, resistant cultivars offered partial protection against WSMV and TriMV. Contrary to the field scenario, however, planting of infected seeds from both resistant cultivars led to lower accumulation of WSMV, but not TriMV, in vertically infected plants. Also, TriMV accumulation in progeny plants did not differ between resistant (BT and Joe) and susceptible (TAM 304) cultivars. Furthermore, BT seeds with mixed infection had more viruses compared to the Joe seeds with single infection. These disparities in virus accumulation fields and greenhouse experiments are likely due to complex interactions between viruses, resistance genes, and the environment (temperature), which need to be dissected further at the molecular level.

Our findings reveal that seeds from plants with mixed infection can carry both WSMV and TriMV in low frequency but may result in the early establishment of mixed infections in next-generation plants. Although WSMV and TriMV accumulation did not differ between single or mixed-infected seeds, mixed-infected plants from these seeds produced severe disease phenotypes. Previous studies have also shown that WSMV and TriMV act synergistically in co-infected plants resulting in increased disease severity and yield losses [6,7,8]. In many systems, seed transmission of viruses acting as a primary source of infection can have significant epidemiological significance, where vectors can quickly acquire the virus resulting in rapid transmission within homogenous fields [37]. This is a strong possibility in wheat fields planted with WSMV-resistant cultivars where few singly or mixed-infected seeds can serve as an initial source of inoculum for wheat curl mites, especially biotype II, which is the most prevalent mite biotype in the Texas Panhandle and is known to efficiently transmit both viruses and to produce severe disease symptoms [38].

WSMV and TriMV accumulation in seeds collected from field-infected plants was on par with WSMV and TriMV accumulation initially observed (week 1) in field-infected plants. Theoretically, virus accumulation in these seeds was possibly over the inoculum threshold required to start the infection. However, very few plants from these infected seeds developed disease phenotypes. This could have been because both WSMV and TriMV accumulated in the seed coat, resulting in decreased opportunities for the virus to infect the developing embryo. Transmission rates for WSMV in the current study were higher than previous reports of 0.5–1.5% [26,28]. These discrepancies were most likely due to differences in seed collection methods, wheat genetics, or diagnostic techniques used (qRT-PCR vs. ELISA). Contrary to our study, where seed samples came from plants that were mixed infected with WSMV and TriMV, previous studies either used seeds collected from lots originating from a breeding nursery known to have been infected by WSMV or seeds randomly collected directly after harvest from wheat crops showing WSMV symptoms [26,28]. qRT-PCR was used in the present study to analyze percent infection, as it was a more accurate and highly sensitive technique compared to ELISA [9,39]. Our prior study recommended not using ELISA for studies requiring a high degree of sensitivity and accuracy in virus detection [9]

## 5. Conclusions

Novelty in our study was twofold. For the first time, we reported seed transmission of TriMV individually as well as in the mixed infection with WSMV. Both *Wsm1* and *Wsm2* significantly reduced the vertical transmission of WSMV. However, no such effects were observed on TriMV, especially in progeny plants. Furthermore, the mixed transmission of TriMV with WSMV in the infected progeny plants resulted in a severe WSM disease phenotype. Higher proportions of mixed infection and disease were observed in susceptible cultivars (TAM 304) than in resistant cultivars (BT and Joe), highlighting the importance of host-plant resistance in controlling WSM. Current commercial wheat cultivars carrying WSMV-resistant genes might become vulnerable to WSM due to the mixed infection with TriMV. Early establishment of such a mixed infection vertically via seed may serve as a source of inoculum for horizontal transmission via mites. Therefore, to mitigate the risk posed by TriMV to the wheat industry in Texas and the High Plains, more research is warranted to identify and incorporate host-plant resistance against TriMV.

## Figures and Tables

**Figure 1 viruses-15-01774-f001:**
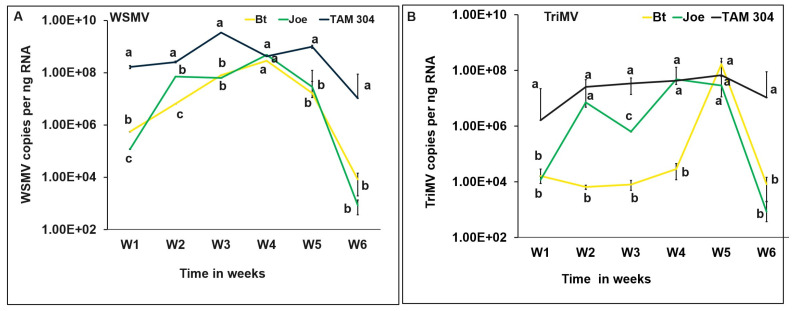
**WSMV and TriMV accumulation in field-infected plants from wheat cultivars with varying genetics (resistant (BT *Wsm1* and Joe *Wsm2*) and susceptible (TAM 304)).** Lines with standard errors represent the average numbers of (**A**) WSMV and (**B**) TriMV copies accumulated in the mixed-infected plants (WSMV and TriMV) over a period of six weeks. Coat protein (CP) gene copy numbers were estimated by qRT-PCR, followed by absolute quantitation using plasmids containing CP gene inserts as standards. The different letters on error bars indicate significant differences between means at α = 0.05. The *Y*-axis has a logarithmic scale.

**Figure 2 viruses-15-01774-f002:**
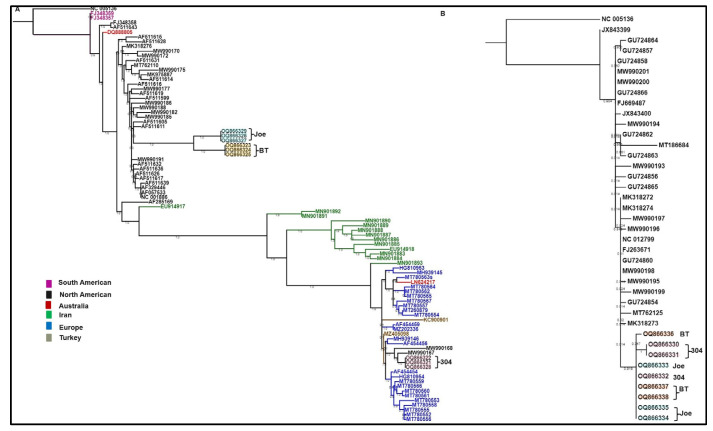
The maximum-likelihood phylogeny of (**A**) WSMV and (**B**) TriMV coat protein gene sequences of Bushland isolates with that of oat necrotic mottle virus (ONMV; NC_005136) as an outgroup. WSMV and TriMV Bushland isolate are highlighted as follows: TAM 304 (pink), Joe (blue), and BT (orange). Support for nodes in a bootstrap analysis with 1000 replicates, which is shown. The branch to the outgroup is drawn as 10% of its actual length.

**Figure 3 viruses-15-01774-f003:**
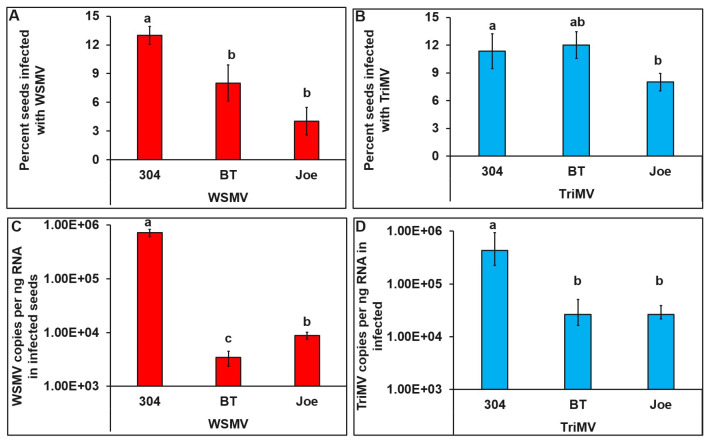
**Single infection of WSMV or TriMV in wheat seeds collected from field-infected plants from cultivars with varying genetics (resistant (BT, *Wsm*1 and Joe *Wsm*2) and susceptible cultivars (TAM 304)).** Bar graphs with standard errors represent the average percentage of seeds that tested positive for (**A**) WSMV and (**B**) TriMV in qRT-PCR analysis. Bar graphs with standard errors represent the average numbers of (**C**) WSMV and (**D**) TriMV copies accumulated in the seeds that tested positive for WSMV or TriMV, respectively. Coat protein (CP) gene copy numbers were estimated by qRT-PCR followed by absolute quantitation using plasmids containing CP gene inserts as standards. The different letters on bar graphs indicate significant differences between means at α = 0.05. For the copy numbers, the *Y*-axis has a logarithmic scale.

**Figure 4 viruses-15-01774-f004:**
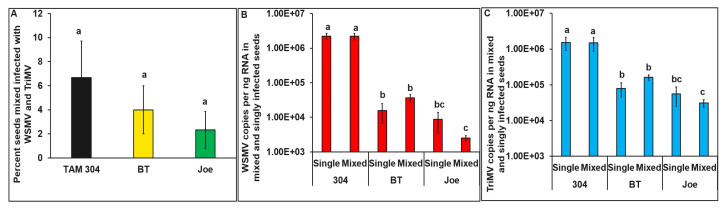
**Mixed infection of WSMV and TriMV in wheat seeds collected from field-infected plants from cultivars with varying genetics (resistant (BT, *Wsm*1 and Joe *Wsm*2) and susceptible cultivars (TAM 304).** (**A**) Bar graphs with standard errors represent the average percentages of seeds that tested positive for both WSMV and TriMV in qRT-PCR analysis (**B**) Bar graphs with standard errors represent the average number of WSMV copies accumulated in the seeds infected singly with WSMV or mixed infected with WSMV and TriMV. (**C**) Bar graphs with standard errors represent the average number of TriMV copies accumulated in the seeds infected singly with TriMV or mixed infected with WSMV and TriMV. Coat protein (CP) gene copy numbers were estimated by qRT-PCR followed by absolute quantitation using plasmids containing CP gene inserts as standards. The different letters on bar graphs indicate significant differences between means at α = 0.05. For the copy numbers, the *Y*-axis is on a logarithmic scale.

**Figure 5 viruses-15-01774-f005:**
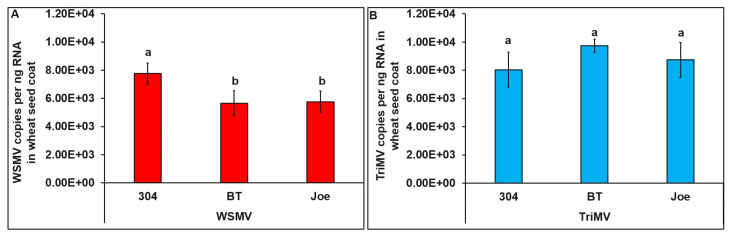
**WSMV or TriMV accumulation in different parts of wheat seeds.** Bar graphs with standard errors represent the average numbers of (**A**) WSMV and (**B**) TriMV copies accumulated in the seed coats of TAM304, BT, and Joe seeds. Coat protein (CP) gene copy numbers were estimated by qRT-PCR, followed by absolute quantitation using plasmids containing CP gene inserts as standards. The different letters on bar graphs indicate significant differences between means at α = 0.05. For the copy numbers, the *Y*-axis has a logarithmic scale.

**Figure 6 viruses-15-01774-f006:**
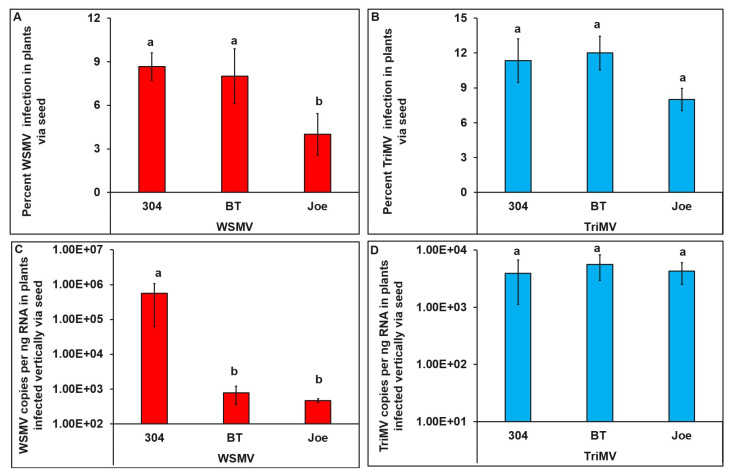
**WSMV or TriMV infection in plant emerged from wheat seeds collected from field-infected plants belonging to cultivars with varying genetics (resistant (BT, *Wsm1* and Joe *Wsm2*) and susceptible cultivars (TAM 304)).** Bar graphs with standard errors represent the average percentages of plants that tested positive for (**A**) WSMV and (**B**) TriMV in qRT-PCR analysis. Bar graphs with standard errors represent the average numbers of (**C**) WSMV and (**D**) TriMV copies accumulated in the plants infected with WSMV or TriMV, respectively. Coat protein (CP) gene copy numbers were estimated by qRT-PCR, followed by absolute quantitation using plasmids containing CP gene segments as standards. The different letters on bar graphs indicate significant differences between means at α = 0.05. For the copy numbers, the *Y*-axis has a logarithmic scale.

**Figure 7 viruses-15-01774-f007:**
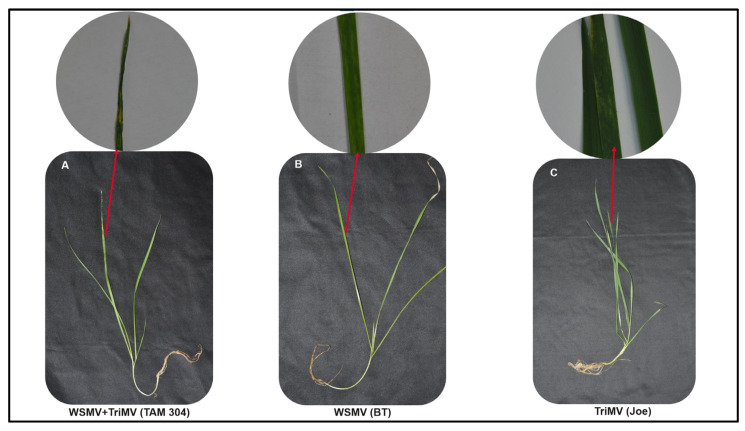
**Wheat streak mosaic phenotype in resistant (BT and Joe) and susceptible TAM 304 plants infected vertically with WSMV and/or TriMV via seeds.** (**A**) TAM 304 plants mixed infected with WSMV and TriMV. (**B**) BT plants infected with WSMV. (**C**) Joe plants infected with TriMV.

## Data Availability

Not applicable.

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
