# Peer review of "Seed Transmission of Wheat Streak Mosaic Virus and Triticum Mosaic Virus in Differentially Resistant Wheat Cultivars"

_viruses, 2023, doi:10.3390/v15081774_

Round 1

Reviewer 1 Report

Gautam et al. describe seed transmission of wheat streak mosaic virus (WSMV) and Triticum mosaic virus (TriMV) from field-collected wheat cultivars that are differentially resistant to these two viruses. The authors found that a relatively high percentage of wheat seeds from infected plants were positive for WSMV or TriMV. In grow-out experiments, the authors found that WSMV seed transmitted at 9%, 3%, and 2% In TAM 304, BT, and Joe cultivars, respectively, compared to 10%, 7%, and 8% seed transmission of TriMV.  The seed transmission rate of WSMV reported by this manuscript is substantially higher compared to 3.1% and 0.6% in Spring and Winter wheat cultivars, respectively, from Montana (Plant Disease, In press) and also published reports from Australia. The seed transmission of TriMV is first-time reported in this manuscript. The main problem with this manuscript is that it is a poorly written manuscript with inaccurate statements, and the conclusions presented in this manuscript are questionable/overstated.

 Major concerns: 

  1. The results presented in this manuscript may have unintended consequences on the recommendation to control wheat streak mosaic disease, an economically important viral disease in the Great Plains region. The design of experiments and conclusions drawn from these experiments do not indicate that care was taken to report important results.
  2. The authors used qRT-PCR to detect viruses in wheat seeds collected from the field, and the highly sensitive nature of the test skewed the results to a higher percentage of infection in the seed and in testing progeny plants in the grow-out test. There is no reason to use this highly sensitive qRT-PCR, and the authors proved that these results are not accurate because only 21 to 30% of progeny plants that tested positive in qRT-PCR showed symptoms in plants.
  3. The authors should test the samples by RT-PCR and DAS-ELISA in the progeny plants. The positive data by PCR or qPCR may not reveal whether the infectious virus is present in those samples. Particularly, ELISA should be performed for progeny plants of grow-out test to inconclusively prove that they are detecting the live virus.
  4. The number of plants tested positive for WSMV and TriMV in 150 seeds collected from field-infected wheat cultivars were presented in histograms with standard error (see Figures 3, 4, and 6). I wonder how you can present SE from just one experiment. 
  5. The authors performed grow-out tests at 18°C and reported infection of BT and Joe cultivars. BT and Joe cultivars harbor Wsm1- and Wsm2-resistant genes, respectively. The Wsm1 and Wsm2 genes provide resistance to WSMV+TriMV and WSMV at 18-20°C. I am surprised how the progeny plants of Joe and BT cultivars were infected with WSMV and WSMV/TrIMV in grow-out tests.
  6. I strongly suggest the authors perform seed transmission tests with seeds collected from wheat plants that are mechanically inoculated under greenhouse conditions. This will increase the confidence level in this publication. 

Abstract:

L15: TriMV is not economically important worldwide.

L16: WSM disease is mainly due to WSMV, but some publications also referred to infection by WSMV, TriMV, and HPWMoV. This is the wrong statement.

L38-39: TriMV has a minor role in WSM disease compared to WSMV. This is a wrong statement. 

Introduction:

L45: Why Texas here?

L64-69: These sentences need re-writing.

L112: A cut-off value of 35 and three ct values below that are considered positive may give false positives.

L142: Change ‘centrifuge tube’ to Eppendorf tube’.

L149-150: ‘from 1011 to 101’ should be the other way round as 10-1 to 10-11.

L167: Why it is ‘Six-weeks post-inoculation…’. It looks like the authors inoculated with viruses. 

Results

L191: BT cultivar is resistant to both WSMV and TriMV.

L220: Seeds infested? What is this?

L215: This is the wrong statement. ONMV was used as an outgroup, but you did not use this for comparison.

L238: These are histograms, not the bars. 

Discussion

L313: 15-23% is wrong. This should be 21-39%.

L319-323: This is a wild guess.

Requires moderate English edits. 

Author Response

Gautam et al. describe seed transmission of wheat streak mosaic virus (WSMV) and Triticum mosaic virus (TriMV) from field-collected wheat cultivars that are differentially resistant to these two viruses. The authors found that a relatively high percentage of wheat seeds from infected plants were positive for WSMV or TriMV. In grow-out experiments, the authors found that WSMV seed transmitted at 9%, 3%, and 2% In TAM 304, BT, and Joe cultivars, respectively, compared to 10%, 7%, and 8% seed transmission of TriMV.  The seed transmission rate of WSMV reported by this manuscript is substantially higher compared to 3.1% and 0.6% in Spring and Winter wheat cultivars, respectively, from Montana (Plant Disease, In press) and also published reports from Australia. The seed transmission of TriMV is first-time reported in this manuscript. The main problem with this manuscript is that it is a poorly written manuscript with inaccurate statements, and the conclusions presented in this manuscript are questionable/overstated.

The key reasons why higher percentage of TriMV and WSMV infections were observed in this study was because we used probe-based qRT-PCR which is very accurate and highly sensitive in detecting plant viral RNA in infected tissues and differences in seed collection methods. We did not consider using ELISA because previously our research group conclusively proved that “… the ELISA test, which is commonly used by diagnostic laboratories in the Great Plains, should not be used for studies requiring a high degree of sensitivity and accuracy in virus detection.” (Bryan et al. 2019). Furthermore, qRT-PCR is 1,000 and 10,000 times more sensitive than DAS-ELISA in detecting viruses in seeds (Torre et al. 2019). We collected seeds from WSM infected plants in the field (which has been made abundantly clear in the abstract, methods and discussion) vs random sampling in prior studies leading to higher seed transmission percentages. Reviewer seems to have clearly missed this point as all his arguments center around using qRT-PCR vs ELISA without due consideration to other factors. The detailed description of both scenarios is provided in the discussion (Lines 391-401).

To confirm the reliability of qRT-PCR results, CP gene of TriMV and WSMV from representative samples (with the highest Ct values) belonging to different cultivars were amplified and sequenced. Given the accuracy and accessibility, it is safe to say that most plant virology labs around the world rely on PCR for virus detection over ELISA because protein-based detection cannot detect low titer of virus in samples. Knowing the amount of virus (through highly sensitive techniques such as qRT-PCR) and its association with disease incidence/severity in the field is critically important to accurately model/predict disease development, generate epidemiological meta data and extend disease management recommendations.

We appreciate the reviewers’ efforts and time in helping us to improve the manuscript. However, the reviewer 1 appears to be generically critical of results because they don’t match with prior studies (from different locations) and of writing without providing specific examples of where and how the quality of writing needs to be improved (manuscript was thoroughly reviewed by native English speakers). The reviewer seems to have a preconceived opinion on the lower seed transmission (3.1 to 0.6%) of WSMV. Higher percentages of seed transmission in our study than previously thought brings new information to light for growers/seed companies/breeders who may wish to consider ways to keep percent infection of viruses in seeds low (which ELISA won’t be able to detect) to minimize potential disease outbreaks in the field. Also, because the seed transmission is likely to be determined by disease incidence/severity in field, environmental parameters, and geographical locations (Texas vs Montana vs Australia), it is no surprise that the WSMV seed transmission percentages were different in different studies (besides the techniques used/seed collection method). Because the reviewer 1 did not take these factors into account, their comments as they mostly criticize the widely used diagnostic technique, appear to be presumptuous.

For the robust justification provided above, we respectfully decline to re-analyze the samples with a less sensitive DAS-ELISA.

Bryan, B.; Paetzold, L.; Workneh, F.; Rush, C.M. Incidence of Mite-Vectored Viruses of Wheat in the Texas High Plains and Interactions With Their Host and Vector. Plant Dis 2019, 103, 2996–3001, doi:10.1094/PDIS-03-19-0620-SR.

Torre, C.; Agüero, J.; Gómez-Aix, C.; Aranda, M.A. Comparison of DAS-ELISA and qRT-PCR for the detection of cucurbit viruses in seeds. Ann. Appl. Biol. 2020, 176, 158–169

Major concerns: 

The results presented in this manuscript may have unintended consequences on the recommendation to control wheat streak mosaic disease, an economically important viral disease in the Great Plains region. The design of experiments and conclusions drawn from these experiments do not indicate that care was taken to report important results.

We understand the reviewers’ concern regarding recommendations for wheat streak mosaic disease in Great Plains. We’ve carefully designed the study, accurately presented the results and conclusions. We certainly hope to have addressed reviewers’ criticism in earlier comments: Knowing the amount of virus (through highly sensitive techniques such as qRT-PCR) and its association with disease incidence/severity in the field is critically important to accurately model/predict disease development, generate epidemiological meta data and extend disease management recommendations. Higher percentages of seed transmission in our study than previously thought brings new information to light for growers/seed companies/breeders who may wish to consider ways to keep percent infection of viruses in seeds low (which ELISA won’t be able to detect) to minimize potential disease outbreaks in the field.

The authors used qRT-PCR to detect viruses in wheat seeds collected from the field, and the highly sensitive nature of the test skewed the results to a higher percentage of infection in the seed and in testing progeny plants in the grow-out test. There is no reason to use this highly sensitive qRT-PCR, and the authors proved that these results are not accurate because only 21 to 30% of progeny plants that tested positive in qRT-PCR showed symptoms in plants.

We do not agree with the reviewers’ comment on not using sensitive qRT-PCR for detecting viruses in samples for the reasons provided above. As stated above, we used qRT-PCR instead of ELISA because previously our research group found that qRT-PCR is very accurate and highly sensitive (Bryan et al. 2019). Disease development in fact is a complex process and requires the favorable interaction of pathogen, host, and environment. Thus, seed transmission doesn’t necessarily lead to measurable disease development because of other factors that come into play after transmission occurred. Across the cultivars, 21-39% of plants that tested positive for WSMV or TriMV showed phenotypic disease expression. On the contrary, 100% of plants that had vertical infection with both viruses via seeds developed WSM disease phenotype. This does not prove that results were inaccurate.

Bryan, B.; Paetzold, L.; Workneh, F.; Rush, C.M. Incidence of Mite-Vectored Viruses of Wheat in the Texas High Plains and Interactions With Their Host and Vector. Plant Dis 2019, 103, 2996–3001, doi:10.1094/PDIS-03-19-0620-SR.

The authors should test the samples by RT-PCR and DAS-ELISA in the progeny plants. The positive data by PCR or qPCR may not reveal whether the infectious virus is present in those samples. Particularly, ELISA should be performed for progeny plants of grow-out test to inconclusively prove that they are detecting the live virus.

Plant diagnostic techniques do not rely on detecting live virus in plants. ELISA most certainly does not detect and ascertain the presence of live virions in plant samples if that’s what reviewer means. qRT-PCR is 1,000 and 10,000 times more sensitive than DAS-ELISA in detecting viruses in seeds (Torre et al. 2019). Therefore, there was no point in doing ELISA on grown-out plants when we could detect viral nucleic acid in samples with a more sensitive and reliable technique.

Torre, C.; Agüero, J.; Gómez-Aix, C.; Aranda, M.A. Comparison of DAS-ELISA and qRT-PCR for the detection of cucurbit viruses in seeds. Ann. Appl. Biol. 2020, 176, 158–169

The number of plants tested positive for WSMV and TriMV in 150 seeds collected from field-infected wheat cultivars were presented in histograms with standard error (see Figures 3, 4, and 6). I wonder how you can present SE from just one experiment. 

Seeds were collected from 15-20 different infected plants from the field and divided into three lots. Each lot contains seeds from 5-7 infected plants. Then 50 seeds from each lot were tested for viruses and the experiment was replicated three times. The same goes for the seeds used in grow-out experiments, where 30 seeds were selected from each lot. The text has been edited to make it clear. Lines116-120, 155-157, 170.

The authors performed grow-out tests at 18°C and reported infection of BT and Joe cultivars. BT and Joe cultivars harbor Wsm1- and Wsm2-resistant genes, respectively. The Wsm1 and Wsm2 genes provide resistance to WSMV+TriMV and WSMV at 18-20°C. I am surprised how the progeny plants of Joe and BT cultivars were infected with WSMV and WSMV/TriMV in grow-out tests.

Yes, Wsm1- and Wsm2- are reported to resist WSM at temperatures ≤18 °C. However, they don’t make plants completely immune. At a temperature of 18 ± 1 °C, wheat can become infected with WSMV and/or TriMV as shown in previous studies (Tatineni et al. 2016). Also, depending on the isolates of WSMV, the disease can develop in resistant plants (Kumssa et al. 2019) at 18 °C. The development of disease depends on complex interactions of pathogen, host, and environment, not plant resistance status alone.

Tatineni, S., Wosula, E. N., Bartels, M., Hein, G. L., and Graybosch, R. A. (2016). Temperature-dependent Wsm1 and Wsm2 gene-specific blockage of viral long-distance transport provides resistance to Wheat streak mosaic virus and Triticum mosaic virus in wheat. Mol. Plant Microbe Interact. 29, 724–738. doi: 10.1094/MPMI-06-16-0110-R

Kumssa, T. T., Rupp, J. S., Fellers, M. C., Fellers, J. P., and Zhang, G. (2019). An isolate of Wheat streak mosaic virus from foxtail overcomes Wsm2 resistance in wheat. Plant Pathol. 68, 783–789. doi: 10.1111/ppa.12989

I strongly suggest the authors perform seed transmission tests with seeds collected from wheat plants that are mechanically inoculated under greenhouse conditions. This will increase the confidence level in this publication. 

We strongly disagree with the reviewer; using mechanical inoculation in a greenhouse to study seed transmission and drawing conclusions on that basis instead of a rigorous field study using highly sensitive technique is both inaccurate, misleading and has “unintended consequences on the recommendation to control wheat streak mosaic disease”. This study was specifically focused on the seed transmission of WSMV and TriMV in mixed-infected plants in the field. For that reason, repeating the whole experiment in the greenhouse via mechanical inoculation is beyond the scope of this manuscript. In general, findings of greenhouse studies are not as robust as field studies.

Abstract:

L15: TriMV is not economically important worldwide.

Worldwide has been removed. Lines15.

L16: WSM disease is mainly due to WSMV, but some publications also referred to infection by WSMV, TriMV, and HPWMoV. This is the wrong statement.

The sentence has been edited. Lines16.

L38-39: TriMV has a minor role in WSM disease compared to WSMV. This is a wrong statement. 

It is true that TriMV was identified much later than WSMV. However, during a recent survey conducted by our research group in Texas High Plains, we found out of 648 samples that tested positive for WSMV, 93% of them were also positive for TriMV (Bryan et al. 2019). Furthermore, recently in Kansas, mixed infection of TriMV with WSMV was found in 8-14% of the wheat streak-infected plants (Redila et al. 2021). Also, mixed infection of TriMV with WSMV leads to severe disease phenotype even in resistant cultivars (Byamukama et al. 2012). Taken together, TriMV seems to have significant effects on WSM.

Byamukama, E.; Tatineni, S.; Hein, G.L.; Graybosch, R.A.; Baenziger, P.S.; French, R.; Wegulo, S.N. Effects of Single and Double Infections of Winter Wheat by Triticum Mosaic Virus and Wheat Streak Mosaic Virus on Yield Determinants. Plant Dis 2012, 96, 859–864, doi:10.1094/PDIS-11-11-0957-RE

Bryan, B.; Paetzold, L.; Workneh, F.; Rush, C.M. Incidence of Mite-Vectored Viruses of Wheat in the Texas High Plains and Interactions With Their Host and Vector. Plant Dis 2019, 103, 2996–3001, doi:10.1094/PDIS-03-19-0620-SR.

Redila CD, Phipps S and Nouri S (2021) Full Genome Evolutionary Studies of Wheat Streak Mosaic Associated Viruses Using High-Throughput Sequencing. Front. Microbiol. 12:699078. doi: 10.3389/fmicb.2021.699078

Introduction:

L45: Why Texas here?

Text has been edited for clarity. Line 46

L64-69: These sentences need re-writing.

Text has been edited for clarity. Lines 66-67.

L112: A cut-off value of 35 and three ct values below that are considered positive may give false positives.

A cut off three Ct values below 35 was decided because in such samples (100%), virus titer was high enough to amplify and sequence CP genes of TriMV and WSMV. Thus, avoiding false positives.

L142: Change ‘centrifuge tube’ to Eppendorf tube’.

We used tubes procured from Fisher Scientific. The text has been edited for the technical name of the tube and the cat# number. Lines 144- 146.

L149-150: ‘from 1011 to 101’ should be the other way round as 10-1 to 10-11.

Text has been edited. Line 154

L167: Why it is ‘Six-weeks post-inoculation…’. It looks like the authors inoculated with viruses. 

Text has been edited for clarity. Line 174.

Results

L191: BT cultivar is resistant to both WSMV and TriMV.

The text has been edited. Lines 198-200.

L220: Seeds infested? What is this?

Text has been edited for clarity. Lines 222.

L215: This is the wrong statement. ONMV was used as an outgroup, but you did not use this for comparison.

Text has been edited to remove the confusion. Lines 238-239.

L238: These are histograms, not the bars. 

Not true. These are indeed bar graphs as they present data on categorical variables. On the contrary histograms present data by way of bars to show the frequency of numerical data. More information here: https://keydifferences.com/difference-between-histogram-and-bar-graph.html#:~:text=Histogram%20refers%20to%20a%20graphical%20representation%3B%20that%20displays%20data%20by,frequency%20distribution%20of%20continuous%20variables.

Discussion

L313: 15-23% is wrong. This should be 21-39%.

The text has been edited as suggested. Line 327.

L319-323: This is a wild guess.

After a literature search, we did not come across any study that investigated the mixed seed transmission of TriMV with WSMV or examined TriMV and WSMV accumulation in wheat seed tissues. If the reviewer can provide examples of such studies, we will acknowledge those studies and alter the text accordingly.  Also, why TriMV is important has been discussed above.

Reviewer 2 Report

There are a lot of new information on two viruses causing wheat streak mosaic disease, especially on TriMV. Considering the invaluable information for both farmers and scientists, I believe that the manuscript deserves to be published in the Viruses journal. Though, I believe that the manuscript could be improved by paying attention to the following issues:

Figure 4B and Figure 5A seem to show inconsistent results. Figure 4B shows that infected seeds of Joe contain less WSMV copies compared to BT, while Figure 5A shows that Joe and BT seeds contain similar levels of WSMV.

It is not easy to understand the statement “100% of plants ….. seed-developed WSM disease phenotype.” (line #314-315). Can we revise the statement to “100% of plants that had vertical infection with both viruses via seed developed WSM disease phenotype.”. And, can we see the data?

In lines #338-339, the authors mentioned about the decline of virus accumulation after initial increase up to four weeks. It is confusing. To me, the statement seems to mean that the decline of the virus accumulation was probably due to resistance breakdown at high temperature. Is it true? If the resistance broke down, it would lead to increase of virus accumulation. Are you trying to say that the resistance breakdown at high temperature probably caused the initial increase of the virus accumulation?

Figure 4 shows that BT seeds with mixed infection had more viruses compared to the seeds with single infection. In the case of Joe seeds, it is the opposite. It would be nice if the authors give an explanation for this interesting result.

The abbreviation BT is explained in line #99. Though, the explanation should have been done earlier (in line #20).

It would be nice for readers if the authors describe how F values were calculated.

In Figure 1, bars are not employed to represent the data. However, in the figure legend, the authors used the expression ‘different letters on bars’ (line #199).

In Figure 2, the letters ‘A’ and ‘B’ indicating the two panels are too small to recognize.

The expression ‘collected seeds’ in line #270 could be misleading. To make it clear what the authors meant, it would be better to replace it with ‘infected seeds’ as in line #275.

English is fine.

Author Response

There are a lot of new information on two viruses causing wheat streak mosaic disease, especially on TriMV. Considering the invaluable information for both farmers and scientists, I believe that the manuscript deserves to be published in the Viruses journal. Though, I believe that the manuscript could be improved by paying attention to the following issues:

We appreciate the reviewers’ efforts and time in helping us to improve the quality of manuscript.

Figure 4B and Figure 5A seem to show inconsistent results. Figure 4B shows that infected seeds of Joe contain less WSMV copies compared to BT, while Figure 5A shows that Joe and BT seeds contain similar levels of WSMV.

Figure 4B is for data generated from the whole seed, while figure 5A shows virus accumulation in the wheat seed coat. Such discrepancy could be the result of differential virus accumulation in wheat seed tissues.

It is not easy to understand the statement “100% of plants ….. seed-developed WSM disease phenotype.” (line #314-315). Can we revise the statement to “100% of plants that had vertical infection with both viruses via seed developed WSM disease phenotype.”. And, can we see the data?

The sentence has been edited as suggested. Data is included in the result section lines: 310-314 and 328-329.

In lines #338-339, the authors mentioned about the decline of virus accumulation after initial increase up to four weeks. It is confusing. To me, the statement seems to mean that the decline of the virus accumulation was probably due to resistance breakdown at high temperature. Is it true? If the resistance broke down, it would lead to increase of virus accumulation. Are you trying to say that the resistance breakdown at high temperature probably caused the initial increase of the virus accumulation?

Yes, WSM-1 and WSM-2 imparted resistance restricts virus movement in the plants. Although the local infection can still take place (Tatineni et al. 2016). At higher temperatures, this resistance breaks down resulting in systemic virus movement, higher accumulation, and increased disease severity in resistant plants.

The text has been edited for clarity. Lines 350-354.

Tatineni, S., Wosula, E. N., Bartels, M., Hein, G. L., and Graybosch, R. A. (2016). Temperature-dependent Wsm1 and Wsm2 gene-specific blockage of viral long-distance transport provides resistance to Wheat streak mosaic virus and Triticum mosaic virus in wheat. Mol. Plant Microbe Interact. 29, 724–738. doi: 10.1094/MPMI-06-16-0110-R

Figure 4 shows that BT seeds with mixed infection had more viruses compared to the seeds with single infection. In the case of Joe seeds, it is the opposite. It would be nice if the authors give an explanation for this interesting result.

WSMV and TriMV accumulation did differ between mixed-infected vs single-infected seeds of BT and Joe cultivars. However, it was not statistically significant.

The abbreviation BT is explained in line #99. Though, the explanation should have been done earlier (in line #20).

The text has been edited as suggested. Line 20.

It would be nice for readers if the authors describe how F values were calculated.

For percent infection data was analyzed with logistic regression and we obtained chi-square values. We apologize for the confusion. F has been replaced with a chi-square symbol in all places.

In Figure 1, bars are not employed to represent the data. However, in the figure legend, the authors used the expression ‘different letters on bars’ (line 199).

Bars have been changed to line. Line 203.

In Figure 2, the letters ‘A’ and ‘B’ indicating the two panels are too small to recognize.

The figure has been edited for clarity. Line 238.

The expression ‘collected seeds’ in line #270 could be misleading. To make it clear what the authors meant, it would be better to replace it with ‘infected seeds’ as in the line 275.

The text has been edited as suggested. Line 284.

Round 2

Reviewer 1 Report

The revised version addressed a few of my concerns but not all. The major flaw of this study, as mentioned in my original review, is developing symptoms in wheat cultivars with Wsm1- or Wsm2-resistant genes at 18C. What is the rationality in incubating seeds at 18C for seed transmission studies? The authors cited Tatinane et al. (2016) for Wsm1- or Wsm2-genes containing wheat cultivars are not immune to viruses. In this published paper, the authors showed that WSMV and TriMV move cell-to-cell but not systemically. In this manuscript, wheat seedlings from BT (Wsm1) and Joe (Wsm2) cultivars developed systemic symptoms at 18C, indicating that the virus moved systemically. The authors cited Kumssa et al. (2019) for infection in resistant wheat cultivars. In this published paper, the authors found Wsm2-resistance-breaking WSMV isolates. Then how about the Wsm1 gene containing MT? It seems the authors misinterpreting published papers for their convenience. If this manuscript is published as is, it is obvious to question how resistant wheat cultivars developed symptoms at 18C. It is up to the handling editor and authors how to fix this problem. I am not trying to block this manuscript from publication.

Minor points:

L15-16: This sentence is not correct. Needs re-writing. 

L20-21: Wsm1 provides resistance to both WSMV and TriMV. 

Author Response

The revised version addressed a few of my concerns but not all. The major flaw of this study, as mentioned in my original review, is developing symptoms in wheat cultivars with Wsm1- or Wsm2-resistant genes at 18C. What is the rationality in incubating seeds at 18C for seed transmission studies? The authors cited Tatinane et al. (2016) for Wsm1- or Wsm2-genes containing wheat cultivars are not immune to viruses. In this published paper, the authors showed that WSMV and TriMV move cell-to-cell but not systemically. In this manuscript, wheat seedlings from BT (Wsm1) and Joe (Wsm2) cultivars developed systemic symptoms at 18C, indicating that the virus moved systemically. The authors cited Kumssa et al. (2019) for infection in resistant wheat cultivars. In this published paper, the authors found Wsm2-resistance-breaking WSMV isolates. Then how about the Wsm1 gene containing MT? It seems the authors misinterpreting published papers for their convenience. If this manuscript is published as is, it is obvious to question how resistant wheat cultivars developed symptoms at 18C. It is up to the handling editor and authors how to fix this problem. I am not trying to block this manuscript from publication.

The key reason why we have grown seeds at 18 oC is to assess WSM symptoms development from putative virus infected seeds in a rigorous manner. We wanted to see whether infected seeds are able to develop disease (express symptoms) phenotype in resistant cultivars at the temperature at which resistance genes are surely expressed, if not best expressed. This is also a congenial temperature for optimal wheat growth. The presence of Wsm1 and Wsm2 genes in resistant cultivars does not eliminate the possibility of disease development. As we explained earlier, disease development depends on complex interaction between host, pathogen, and environment. It is ironic that reviewer 1 is criticizing disease development in resistant hosts at 18 oC in this round (at higher temperatures, the seed transmission percentages would have been possibly even higher because of resistance breaking down), whereas they seemed to be concerned about high seed transmission percentages overall in the first round of revision.

We cited Kumssa et al. 2019 study with reference to differential interactions of WSMV virus isolates with Wsm-2 resistant gene at 18oC. It is unclear if this is true for Wsm-1 and warrants further investigation. We did not claim anywhere in the manuscript that the seed-transmitted viruses establish a systemic infection in resistant plants as it is beyond the scope of the present study. We documented WSMV and TriMV move via seed in resistant plants and in some instances can cause mild disease phenotype (Fig. 7). If it was a localized infection in leaf tissues or a virus did establish a systemic infection in resistant plants at 18 ± 1oC remains to be explored.

So, we’re surely not “misinterpreting published papers for our convenience” but simply reporting what we observed. Because all of reviewer’s comments were justifiable (which we’ve strongly justified), we are not sure why reviewer 1 appeared to exaggerate concerns and insisted on major revisions at both occasions.

Minor points:

L15-16: This sentence is not correct. Needs re-writing. 

Sentence edited for clarity.

L20-21: Wsm1 provides resistance to both WSMV and TriMV. 

Edited as suggested.